# Antimicrobial Activity of Green Silver Nanoparticles Synthesized by Different Extracts from the Leaves of Saudi Palm Tree (*Phoenix Dactylifera* L.)

**DOI:** 10.3390/molecules27103113

**Published:** 2022-05-12

**Authors:** Jihan F. Al Mutairi, Fatimah Al-Otibi, Hassna M. Alhajri, Raedah I. Alharbi, Saud Alarifi, Seham S. Alterary

**Affiliations:** 1Chemistry Department, College of Science, King Saud University, P.O. Box 22452, Riyadh 11495, Saudi Arabia; jehanfeisal@gmail.com (J.F.A.M.); 441203417@student.ksu.edu.sa (H.M.A.); salterary@ksu.edu.sa (S.S.A.); 2Department of Botany and Microbiology, College of Science, King Saud University, P.O. Box 22452, Riyadh 11495, Saudi Arabia; raalharbi@ksu.edu.sa; 3Zoology Department, College of Science, King Saud University, P.O. Box 22452, Riyadh 11495, Saudi Arabia; salarifi@ksu.edu.sa

**Keywords:** *Phoenix dactylifera* L., silver nanoparticles, *Candida*, antimicrobial activities

## Abstract

The Arabian desert is rich in different species of medicinal plants, which approved variable antimicrobial activities. *Phoenix dactylifera* L. is one of the medical trees rich in phenolic acids and flavonoids. The current study aimed to assess the antibacterial and antifungal properties of the silver nanoparticles (AgNPs) green-synthesized by two preparations (ethanolic and water extracts) from palm leaves. The characteristics of the produced AgNPs were tested by UV-visible spectroscopy and Transmitted Electron Microscopy (TEM). The antifungal activity of *Phoenix dactylifera* L. was tested against different species of *Candida*. Moreover, its antibacterial activity was evaluated against two Gram-positive and two Gram-negative strains. The results showed that AgNPs had a spherical larger shape than the crude extracts. AgNPs, from both preparations, had significant antimicrobial effects. The water extract had slightly higher antimicrobial activity than the ethanolic extract, as it induced more inhibitory effects against all species. That suggests the possible use of palm leaf extracts against different pathogenic bacteria and fungi instead of chemical compounds, which had economic and health benefits.

## 1. Introduction

*Phoenix dactylifera* L., or a palm tree, is a member of the Palmae family and is considered one of the most important and economic plants in the Arab world [1]. These trees had spiritual and cultural influences on the human population of the Arabian Peninsula [2]. The products of palm trees had medical importance because of the high phenolic and flavonoids content [3,4]. Different countries of the Middle East used palm leaves in folk medicine, particularly for pathogenic microbial diseases [5]. Furthermore, the phytochemical screening of palm leaves revealed the presence of different compounds of antimicrobial characteristics, such as carbohydrates, alkaloids, steroids, flavonoids, and tannins [6].

Nanomedicine is one of the recent emerging fields, which involved the usage of nanoparticles as drug delivery systems to potentiate the therapeutic action of a phytochemical agent [7]. Among all used nanoparticles, the noble silver nanoparticles (AgNPs) gain great interest due to their wide applications in the medical field [8,9]. Recently, many efforts were made to synthesize noble metal was AgNPs plant extracts as biogenic nanoparticles [10]. A previous Iranian study used the date palm water extract for the biological synthesis of AgNPs and revealed significant inhibitory effects on *Rhizoctonia solani* (AG2_2) cultures [11]. The Iranian study showed that the 25 µg/mL of the synthesized AgNPs eliminated 83% of the fungal growth [11]. Moreover, 1.56 and 3.12 µg/mL of the synthesized palm AgNPs were the minimum inhibitory concentration and minimum bactericide concentration against *Klebsiella pneumonia* (PCI 602) and *Acinetobacter baumannii* (ATCC 19606), respectively [11]. The efficacy of the antimicrobial activity of AgNPs synthesized by *Phoenix dactylifera* extracts against *Staphylococcus aureus*, *Pseudomonas aeruginosa*, and *Escherichia coli* revealed significant antibacterial ability (10–32 mm diameter), as well [12]. The innovation was the green synthesis of simple and cost-effective AgNPs, which provided stable nanoparticles, and could be an alternative for the large-scale synthesis of AgNPs [13]. Some previous studies indicated the long-term stability of green AgNPs against algae such as *Parachlorella kessleri* [14], or Bactria such as *Bacillus licheniformis* [15]. This indicated that the larger sizes of AgNPs are responsible for their antimicrobial performance [16,17]. Furthermore, the high safety level of the green synthesized nanoparticles, with successful antimicrobial effects, might reduce the excessive usage of antibiotics and antifungal treatments [18].

The current work aimed to evaluate the antibacterial and antifungal properties of the AgNPs synthesized from ethanolic and water extracts from the palm leaves (*Phoenix dactylifera* L.) growing in the Saudi desert. Furthermore, the study aimed to test the capability of the natural green synthesized AgNPs from the crude extracts of palm leaves, which might suggest its medical applications against different microbes. To our knowledge, this is the first trial to prove the ability of Saudi palm leaf extracts ingredients against different pathogens.

## 2. Results and Discussion

### 2.1. Structural Characteristics of the AgNPs Synthesized from Different Palm Leaves Extracts

The biogenic synthesis of metallic nanoparticles from natural green materials became one of the promising substitutes to the traditional chemical methods. Nanoparticles are characterized by their small sizes and higher surface charges, which enable the adherence of different biogenic materials and allow their efficient delivery to target cells [19]. Silver nanoparticles synthesized from many plants possessed significant antimicrobial activities such as the AgNPs of leaves of *Catharanthus roseus* against *Plasmodium falciparum* [20] and antibacterial activities of AgNPs synthesized by *Clitoria ternatea* and *Solanum nigrum* against *B. subtilis*, *S. aureus*, *S. pyogenes*, *E. coli*, *P. aeruginosa*, and *K. aerogenes* [21].

In the current study, the biogenic synthesis of AgNPs was performed by two extracts from the leaves of *Phoenix dactylifera* L. AgNPs were synthesized by mixing either the water or the ethanolic extracts of *Phoenix dactylifera* L. leaves with AgNO_3_ solution. The colors of both preparations turned dark brown which indicated the reduction in Ag^+^ and the synthesis of AgNPs. Similarly, a previous study showed that the aqueous extract of the petiole wood of date palm tree turned into dark brown for the synthesized AgNPs, which was confirmed by UV visible spectrometry, FTIR, X-ray diffraction, and TEM [22].

In the current study, the signature characteristics of the synthesized AgNPs were confirmed by UV visible spectroscopy (Figure 1). The absorption peak was 410–470 mm, which confirmed the AgNPs synthesis. Both extracts showed sharp absorption peaks at 300 nm wavelength (3 a.u), whereas AgNPs showed broad absorption peaks. As shown in Figure 1A, the absorption peak of the aqueous extract dropped and stabilized at 429 nm, whereas the water AgNPs showed a lower peak at 400–500 nm (2.2 a.u) than the crude extract.

That might be because the increase in the particle size and the broad surface plasmon resonance (SPR) peak at 429 nm suggests that the AgNPs were dispersed in the aqueous solution. For the ethanolic extracts, Figure 1B showed that either crude extract or synthesized AgNPs produced three sharp absorption peaks at 300, 429, and 700 nm, besides one broad beak at 200–250 nm. However, the ethanolic AgNPs had higher absorbance (4 a.u) at 300 nm. This indicated the formation of AgNPs with larger sizes that were, unlike water AgNPs, not dispersed. In agreement with our findings, a previous study of Ashraf et al., 2016 showed that AgNPs (20 μM) were recorded in the wavelength range of 200–250 nm [23].

Furthermore, the TEM analysis showed that AgNPs, produced by the aqueous extract, were spherical with an average size of 40–50 nm. The synthesized AgNPs aggregated and clumped 24 h post-incubation (Figure 2). For accurate comparison, we have to stain all slides with uranyl acetate and lead citrate to increase the contrast of the ultrastructure. In agreement with our findings, a previous study showed that the green synthesis of AgNPs by *Pedalium murex* leaf extract had higher absorption peaks at 400–450 nm at different concentrations, whereas the TEM showed AgNPs with spherical shapes [24]. Similarly, other studies showed that the biosynthesized AgNPs created from other plants such as *Bidens pilosa* L. [25], *Sabal blackburniana* [26], and *Aaronsohnia factorovskyi* [27] had specific absorption for the synthesized nanoparticles. Also. The TEM results revealed that the produced AgNPs had larger sizes than AgNO_3_ and crude extracts of these plants [25,26,27]. That might be because of the production of free vibrating electrons upon the reduction in Ag^+^ ions, and that causes the formation of the SPR absorption band [25,28]. Several other studies with different plants had similar results [20,21,29,30].

### 2.2. The Phenolic Constituents of Palm Leave Extracts

GC/MS is one of the recent screening tools to identify the chemical composition or the constituents of a specific drug or treatment and its co-metabolites which might explain particular functions or activities [31]. In the current study, we tested the phenolic composition in both extracts of *Phoenix dactylifera* L. leaves (Table 1, Figure 3). GC/MS Analysis of the total phenolic constituents might increase the accuracy of the results as a result of increased stability. The standards database of the NIST libraries, https://www.nist.gov/srd/nist-standard-reference-database-1a (accessed on 10 March 2021), was used to identify the Phyto-compounds and the interpretation of the mass spectrum. GC/MS analysis showed a variety of flavonoids (such as Quercetin, Rutin, and Luteolin) and organic acids (such as p-Coumaric, Caffeic, Chlorogenic, Gallic, Vanillic, Syringic, and Ferulic Acids) [32]. Some of these organic acids are known antioxidants such as Caffeic, p-Coumaric, Chlorogenic, and Ferulic Acids [33]. As shown in Table 1, both extracts had almost similar MS fragmentation in which all compounds had portions of the tested samples. Previous studies showed that these flavonoids and organic acids proved significant antimicrobial activity against dental plaque [34], *Bacillus cereus*, *Salmonella enteritidis*, and *S. aureus* [35]. Similarly, the study of Al-Otibi et al. (2020) showed that the GC/MS analysis revealed many phenolic constituents of *Aaronsohnia factorovskyi*, which had significant antimicrobial activities against *Staphylococcus aureus* and *Fusarium solani* [27]. The strong lipophilicity of the natural plant flavonoids increases the permeability of the host cells, which further increases antimicrobial activity [36].

### 2.3. AgNPs Synthesized by Phoenix dactylifera L. Leaves Extract Had Antifungal Activities

In the current study, the antifungal activities of different concentrations of AgNPs, synthesized by either the water or ethanolic extracts of *Phoenix dactylifera* L. leaves, were tested against *Candida* spp. (Figure 4). The effects of the crude extract and AgNO_3_ were tested, as well. By measuring the zone of inhibition (mm), the results showed that different concentrations of AgNPs had significant inhibitory effects than the crude extract and AgNO_3_ (Table 2 and Table 3). The results revealed that aqueous extract had higher antifungal activity against the pathogenic fungi than the ethanolic extract, and 100% concentration of AgNPs showed the highest significant inhibition against all species (Table 2). The results showed that the *C. albicans* and *C. parapsilosis* were the species most affected by the aqueous extract followed by *C. tropicalis*, where *C. krusei* were the most resistant fungus (Table 2). Unlike aqueous extracts, the ethanolic extracts showed the maximum inhibitory effects against *C. parapsilosis*, *C. tropicalis*, and *C. albicans* (Table 3). The crude ethanolic extract of *Phoenix dactylifera* L. leaves did not induce any antifungal activities against *C. krusei*, however, the biosynthesized AgNPs induced significant inhibitory effects at all concentrations (Table 3). That indicated the ability of the biosynthesize AgNPs to eliminate the pathogenic fungi, even at the lowest concentration.

Furthermore, the TEM imaging of the *C. albicans* treated with both AgNPs preparations showed dramatic structural changes in organelles, accompanied by cell wall rupture in most observed fields. These alterations might have resulted from the action of AgNPs on the genetic material, which decreased the cell density as fewer cells were detected within the microscope’s field, compared to the untreated controls (Figure 5). As seen in the treated fungus was smaller in size than the untreated control. We used higher magnification to show more cellular details indicated by the arrows. The arrows in control image indicated the normal morphology of *C. albicans* such as the cell wall and intracellular organelles. In Figure 5B,C, the arrows indicated the rapture of the cellular membrane, the shrinkage of the cells, without clear distinguishing of the cellular organelles.

Following our findings, multiple previous studies reported the antifungal activities of date palm tree products against different pathogenic fungi. A study from Algeria showed that 90 μg/mL organic extract of pollens of *Phoenix dactylifera* L. had significant antifungal activity against *Fusarium* spp. and *C. albicans* [37].

Another study showed that the aqueous extract of *Phoenix dactylifera* L. fruit could enhance the effects of the known fungicidal “Amphotericin B” against *C. albicans* [38]. Different studies reported the anticandidal activity of AgNPs, biosynthesized by different plant parts of *Phoenix dactylifera* L. [39,40]. Other studies confirmed that silver nanoparticles synthesized from extracts from *Bidens pilosa* L. [25], *Aaronsohnia factorovskyi* [27], and *Senna alexandrina* [41] showed significant antimicrobial activities. To our knowledge, there are no studies on the antifungal effects of *Phoenix dactylifera* L. leaves against other *Candida* spp.

### 2.4. The Bactericidal Activities of AgNPs of Phoenix dactylifera L. Leave Extracts

In the current study, we tested the antibacterial activity of the biosynthesized AgNPs against Gram-positive and Gram-negative bacterial strains. As shown in Figure 6, both the water and ethanolic AgNPs induced a clear inhibition of all strains which gradually increased by increasing the concentration. Neither of the crude extracts nor the AgNO_3_ treatment induced any inhibition of the bacterial growth (data not shown). AgNPs of the aqueous extract had significant inhibition against all species (Table 4). *S. aureus* was the most sensitive strain followed by *P. aeruginosa* (Table 4). Similarly, AgNPs prepared by the ethanolic extract showed higher antibacterial activities of *P. aeruginosa* followed by *S. aureus* (Table 5) than the crude extract. Comparing the two extracts, it is obvious that the AgNPs of the aqueous extract had higher antibacterial activities than AgNPs prepared by the ethanolic extract.

To explore the morphological changes induced by the two AgNPs, we used TEM imaging on *P. aeruginosa* treated by both preparations against an untreated control. TEM revealed that most of the treated cells transformed from smooth rod-shaped morphology to disintegrated cells with broken/damaged membranes, which induced the leakage of cytoplasmic contents to produce ghost cells (Figure 7). We used higher magnification to show more cellular details indicated by the arrows. The arrows in control image indicated the normal morphology of *P. aeruginosa* such as the cell wall and intracellular organelles. In Figure 7B,C, the arrows indicated the rapture of the cellular membrane (pale color) and the damage of the cellular organelles.

Several studies demonstrated the antibacterial activities of *Phoenix dactylifera* L. A previous study showed that the leave extract of *Phoenix dactylifera* had significant antibacterial activity against *Klebsiella pneumoniae* and *Salmonella typhi* [42]. Another study showed that AgNPs from the aqueous root extract of *Phoenix dactylifera* induced a 22 mm zone of inhibition of *E. coli* [39]. Other studies showed that AgNPs of *Phoenix dactylifera* induced significant inhibition zones for different bacterial strains, such as Methicillin-resistant *Staphylococcus aureus* (MRSA) strain [43], *E. coli* [44], *Enterococcus faecalis* [45], and *Listeria monocytogenes* [46].

## 3. Materials and Methods

### 3.1. Plant Material

The date palm tree is one of the famous plants in the Arabian desert. Taxonomic classification of a date palm tree or *Phoenix dactylifera* L. indicated that it belongs to family *Arecaceae*, Genus *Phoenix* L. [47].

In the current study, the fresh green leaves of date palm (*Phoenix dactylifera* L.) were collected, in November 2020 from Al-Medina Region, Saudi Arabia. They were washed and air-dried for two days at room temperature. The dried leaves were ground into fine powder. The bioactive materials were prepared by either water or ethanolic extracts of the dried powder of palm leaves, as described before [47]. Briefly, 30 g of the dried powder were soaked in 300 mL of either distilled water or absolute ethanol (Sigma Aldrich, St. Louis, MS, USA), for 24 h at 4 °C. Both extracts were filtered and incubated at 4 °C until use.

### 3.2. Microorganisms

In the current study, four bacterial strains; *Staphylococcus aureus* (S. aureus) [ATCC 43300], *Bacillus subtilis* (B. subtilis) [ATCC 6051-U] (Gram-positive)*, Pseudomonas aeruginosa* (P. aeruginosa) [ATCC 27852], and Escherichia coli (E. coli) [ATCC 25992] (Gram-negative) (ATCC, Manassas, VA, USA) were provided by King Khaled University Hospital and were used for antibacterial studies, as described before [48]. Five *Candida* species were used in the antifungal studies, as well. The pathogenic fungal species involved *Candida albicans* (*C. albicans*) [ATCC 02091], *Candida tropicalis* (*C. tropicalis*) [ATCC 4563] (ATCC, Manassas, VA, USA), *Candida parapsilosis* (*C. parapsilosis*) [CCRC 20515] (CCRC, Hsinchu, Taiwan), *Candida glabrata* (*C. glabrata*) [CBS 138 (ATCC2001)], and *Candida krusei* (*C. krusei*) [ST-112(ATCC2001)] (ATCC, Manassas, VA, USA). The tested microorganisms were identified by either the Plant Protection Department (Food and Agricultural Sciences College, King Saud University) or the National Center for Research on Agriculture and Livestock, Riyadh, Saudi Arabia.

### 3.3. The Green Synthesis of AgNPs

An amount of 1 mL of each extract was mixed with 10 mL of 2 mM AgNO_3_ (Sigma Aldrich, St. Louis, MS, USA) solution. The mixtures were incubated in dark chambers at room temperature. The color of the mixture changed from yellow (for aqueous extract) or light green (ethanolic extract) to dark brown, which indicates the formation of AgNPs [25]. The synthesized particles were purified by the Size Exclusion Chromatography (SEC) method. The UV-visible absorption spectra and TEM were used to confirm the characteristics of the synthesized AgNPs, which had been described before [26].

### 3.4. Antimicrobial Experiments

The antibacterial properties of the studied materials were evaluated by the Kirby-Bauer disc diffusion method, as was described before [41]. Briefly, the bacterial strains were cultured on Mueller Hinton Agar (MHA) Plates (Sigma Aldrich, St. Louis, MS, USA) for 24 h and incubated at 37 °C. On the next day, 2–3 colonies from each strain were collected and added to a tube of 10 mL distilled water, then 0.2 mL of each bacterial strain (2.5 × 10^5^ CFU/mL) were spread on new MHA plates in a zigzag shape. In each Petri dish, five holes were made with the assistance of a metal cork. The synthesized palm AgNPs (25%, 50%, and 100%), AgNO_3_, and palm crude extract were added, separately, to the holes. After incubation for another 24 h at 37 °C, the inhibition zones surrounding each hole were measured, as described before [41]. The zone of inhibition was computationally calculated by ImageJ version 1.51j8 (National Institutes of Health (NIH), MD, USA).

Regarding the antifungal experiments, the agar well diffusion method was used as been described before [27,41]. Fungi (1 × 10^3^ CFU/mL) were grown on Sabaroud Dextrose Agar Plates (Thermo Fisher Scientific, Waltham, MA, USA) and incubated for 24 h at 37 °C. Similar to bacteria, the strains of Candida were spread on each plate, incubated for 18–24 h at 37 °C, then the mycelial growth was measured in mm. All experiments were performed in triplicates.

### 3.5. Characterization of Synthesized AgNPs

#### 3.5.1. UV Visible Spectroscopy

Both the aqueous and the ethanol extracts were measured on a UV-2450 double-beam UV-Spectrophotometer (Shimadzu, Tokyo, Japan) at 195–950 nm [29]. The measurements were repeated three times for statistical purposes.

#### 3.5.2. Gas Chromatography/Mass Spectrometry Technique (GC/MS)

The GC/MS analysis was performed as was described before [30]. The Clarus 500 Gas Chromatograph/Mass Spectrometer (PerkinElmer, Waltham, MS, USA) was used according to the manufacturer’s instructions. The amount of 2 µL of each extract was injected into the instrument. The experiment was performed in triplicates.

#### 3.5.3. Transmission Electron Microscopy (TEM)

TEM was used to identify the structural characteristics of the synthesized nanoparticles from both extracts of the palm tree. Furthermore, TEM was used also to visualize the morphological changes induced by AgNPs from both extracts in two examples of the studied bacterial or fungal strains. The imaging process was according to the method described [30]. Briefly, after the second fixation with Osmium tetroxide (OsO4), the slides were embedded by a resin mixture using the SPI-PonTM-Araldite^®^ Epoxy Embedding Kit (Structure Probe, Inc. and SPI Supplies, West Chester, PA, USA). Later, the specimen blocks wells were cut into ultrathin sections (70–80 mm) by Leica EM UC6 ultra-microtome (Leica Microsystems, Wetzlar, Germany) and loaded on carbon-coated copper TEM grids. Then, the ultrathin sections were stained by Uranyl Acetate (UA) and Lead Citrate (LC) and examined by JEM-1011 transmission electron microscope (JEOL Ltd. Inc., Tokyo, Japan) at different magnification powers.

### 3.6. Statistical Analysis

All of the experiments were repeated in triplicates for statistical analysis purposes. The statistical analysis by IBM SPSS Statistics software version 21 (IBM, Armonk, NY, USA) was performed. The results were reported as means ± standard deviation (SD). One-way ANOVA was used to assess the significance levels of results at *p* < 0.05. The statistical analysis by One-Way ANOVA, Post-hoc multiple Comparison, Dunnett T3 pattern against mean values of the Crude extract were used to compare different groups of treatments with a sample size (number of replicates) (*n* = 3) and degree of freedom (df) = 2.

## 4. Conclusions

Results of the current study clarify the ability of Saudi palm leaf extract and biosynthesized silver nanoparticles to eliminate pathogenic microbes. That is a readiness to use palm leaf extracts against pathogenic bacteria and fungi. That might aid in maintaining personal health at a lower cost. The antimicrobial efficacy of AgNPs synthesized from natural crude extracts from palm leaves might suggest its possible uses in medical applications. More studies are required to investigate the other antimicrobial activities of different parts of palm trees in vivo.

## Figures and Tables

**Figure 1 molecules-27-03113-f001:**
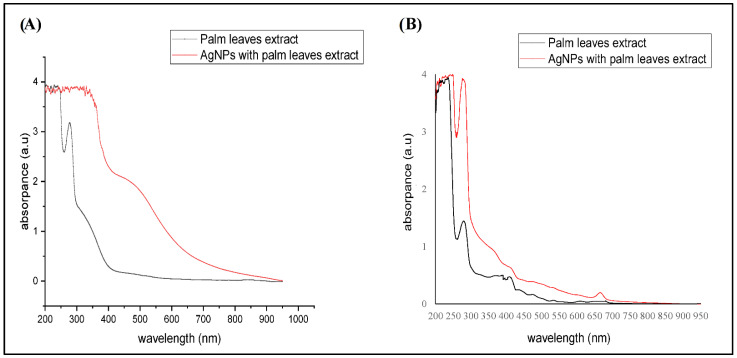
UV-visible spectroscopy of the studied materials. (**A**) Aqueous extract preparations of either the crude extract or biosynthesized AgNPs, (**B**) Ethanolic crude extract and biosynthesized AgNPs.

**Figure 2 molecules-27-03113-f002:**
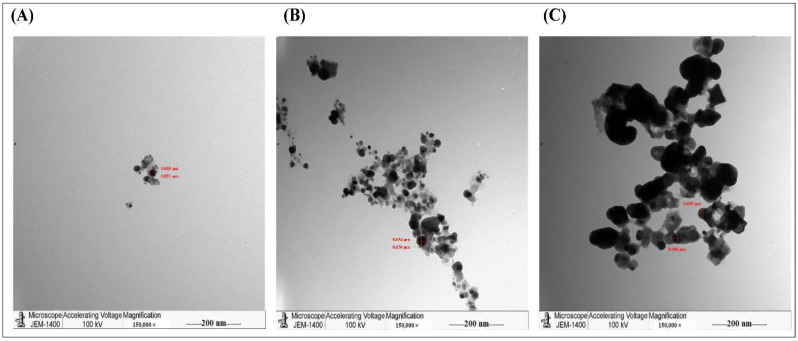
TEM images of the purified AgNPs. It was stained by UA/ LC and examined by the JEM-1011 transmission electron microscope. (**A**) AgNO_3_, (**B**) Aqueous of extract *Phoenix dactylifera* L. leaves, (**C**) AgNPs of *Phoenix dactylifera* L. leaves aqueous extract.

**Figure 3 molecules-27-03113-f003:**
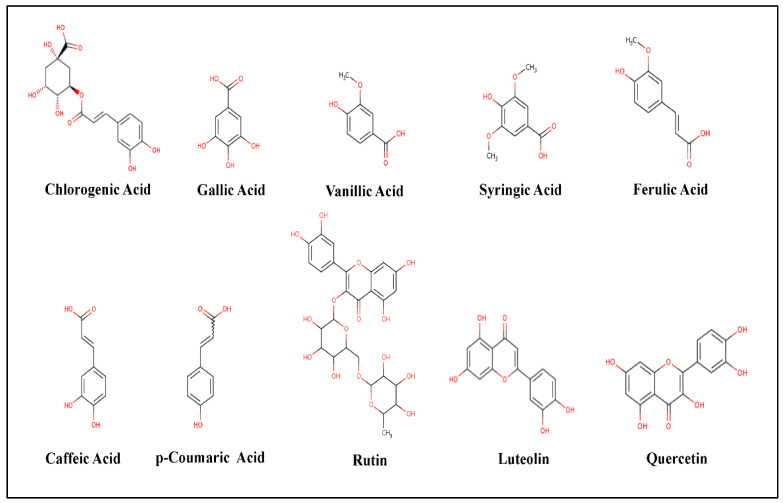
Chemical structures of the phenolic compounds resulted from GC/MS analysis. The chemical structures were created by MolView software (version 2.4) https://molview.org (accessed on 1 February 2020).

**Figure 4 molecules-27-03113-f004:**
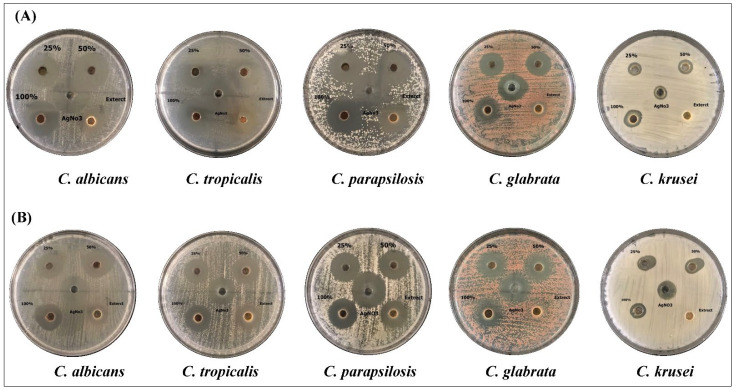
The antifungal activity of AgNPs preparations from *Phoenix dactylifera* L leaf extracts. *Candida* spp. were grown on Sabaroud Dextrose Agar Plates and treated with three concentrations of AgNPs (25, 50, 100% (10 mg/mL), AgNO_3_ (2 mM), and crude extracts (10 mg/mL). (**A**) aqueous extract, (**B**) ethanolic extract.

**Figure 5 molecules-27-03113-f005:**
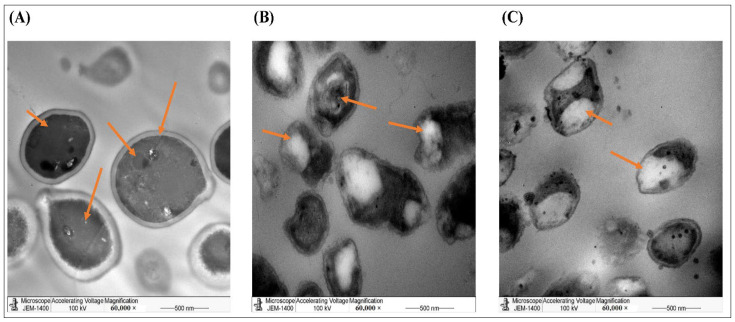
TEM imaging of *C. albicans* showing the morphological changes with different treatments *C. albicans* cells were either (**A**) non-treated (Control), (**B**) treated with AgNPs of the aqueous extract (10 mg/mL), or (**C**) treated with AgNPs of the ethanolic extract (10 mg/mL). Arrows indicated some morphological differences.

**Figure 6 molecules-27-03113-f006:**
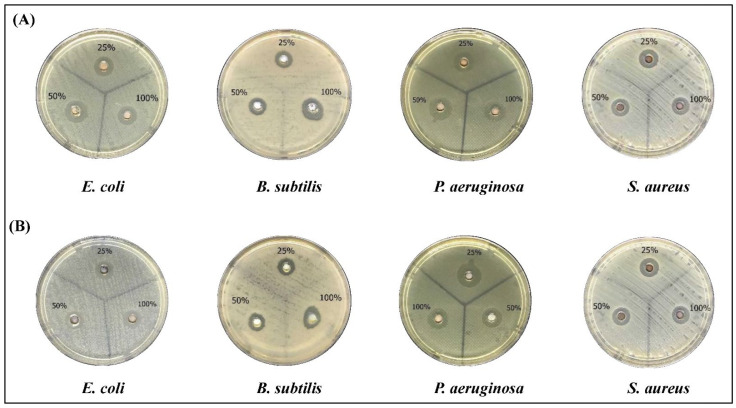
The antibacterial activity of AgNPs preparations from *Phoenix dactylifera* L leaves extracts. Different bacterial strains were grown on Mueller Hinton Agar (MHA) Plates and treated with three concentrations of AgNPs (25, 50, 100% (10 mg/mL)). (**A**) aqueous extract, (**B**) ethanolic extract.

**Figure 7 molecules-27-03113-f007:**
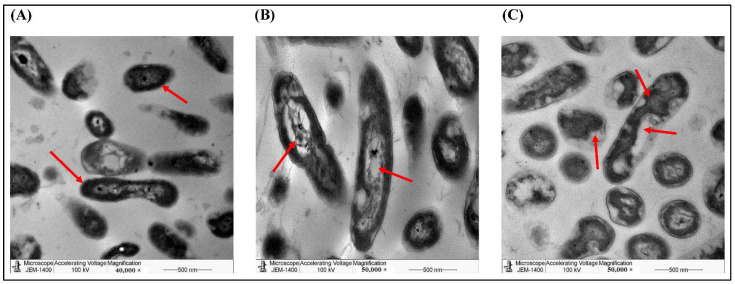
TEM imaging of *P. aeruginosa* showing the morphological changes with different treatments. *P. aeruginosa* cells were either (**A**) non-treated (Control), (**B**) treated with AgNPs of the aqueous extract (10 mg/mL), or (**C**) treated with AgNPs of the ethanolic extract (10 mg/mL). Arrows indicated some morphological differences.

**Table 1 molecules-27-03113-t001:** GC/MS analysis of Phenolic constituents of the Palm leaves extracts.

Phenolic Compounds	Molecular Formula	Molecular Weight g/mol	MS Fragments (*m*/*z*)	
			Aqueous	Ethanol
Chlorogenic Acid	C_16_H_18_O_9_	354.311	353.90	351.00
Gallic Acid	C_6_H_2_(OH)_3_COOH	170.12	169.20	170.10
Vanillic Acid	C_8_H_8_O_4_	168.148	168.20	169.10
Syringic Acid	C_9_H_10_O_5_	198.174	198.20	198.00
Ferulic Acid	C_10_H_10_O_4_	194.186	193.90	195.10
Caffeic Acid	C_9_H_8_O_4_	180.159	179.10	179.10
p-Coumaric Acid	C_9_H_8_O	164.16	166.00	166.10
Rutin	C_27_H_30_O_16_	610.521	606.70	613.10
Luteolin	C_15_H_10_O_6_	286.239	286.00	286.00
Quercetin	C_15_H_10_O_7_	302.238	303.10	303.00

**Table 2 molecules-27-03113-t002:** Zone of inhibition for *Candida* spp. treated with AgNPs of the Palm leaves aqueous extract.

Species		Crude Extract (10 mg/mL)	AgNO_3_ (2 mM)	AgNPs (25%)	AgNPs (50%)	AgNPs (100%)
*C. albicans*	Mean ± SD	20.4 ± 0.2	24.1 ± 0.1	23.3 ± 0.6	24.2 ± 0.2	26.2 ± 0.2
	*p* value #	-------	0.001	0.038	1.4 × 10^−4^	3.7 × 10^−5^
*C. tropicalis*	Mean ± SD	16.3 ± 0.2	23.4 ± 0.2	23.3 ± 0.3	25.0 ± 0.1	27.1 ± 0.2
	*p* value #	-------	3 × 10^−6^	1.1 × 10^−4^	6.4 × 10^−5^	1 × 10^−6^
*C. parapslosis*	Mean ± SD	20.4 ± 0.3	22.1 ± 0.2	22.3 ± 0.3	24.3 ± 0.1	26.2 ± 0.2
	*p* value #	-------	0.008	0.006	0.004	8.1 × 10^−5^
*C. glabrata*	Mean ± SD	10.3 ± 0.1	18.0 ± 0.2	15.0 ± 0.2	17.0 ± 0.1	19.3 ± 0.2
	*p* value #	-------	8.6 × 10^−5^	3 × 10^−4^	6 × 10^−6^	5.9 × 10^−5^
*C. krusei*	Mean ± SD	8.0 ± 0.2	8.9 ± 0.2	10.5 ± 0.5	11.1 ± 0.1	14.4 ± 0.1
	*p* value #	-------	0.022	0.034	1.6 × 10^−4^	7 × 10^−6^

**#** Statistical analysis by One-Way ANOVA, Post hoc multiple Comparison, Dunnett T3 pattern against mean values of the Crude extract, number of replicates (*n* = 3), and Degree of freedom (df) = 2; SD: Standard Deviation.

**Table 3 molecules-27-03113-t003:** Zone of inhibition for *Candida* spp. treated with AgNPs of the Palm leaves ethanolic extract.

Species		Crude Extract (10 mg/mL)	AgNO_3_ (2 mM)	AgNPs (25%)	AgNPs (50%)	AgNPs (100%)
*C. albicans*	Mean ± SD	14.7 ± 0.1	18.1 ± 0.1	18.1 ± 0.1	19.3 ± 0.2	20.3 ± 0.2
	*p* value #	-------	2 × 10^−5^	2 × 10^−5^	2.5 × 10^−5^	1.2 × 10^−5^
*C. tropicalis*	Mean ± SD	17.1 ± 0.1	17.9 ± 0.2	18.3 ± 0.1	19.0 ± 0.1	20.4 ± 0.2
	*p* value #	-------	0.018	0.001	1.3 × 10^−4^	1.4 × 10^−4^
*C. parapslosis*	Mean ± SD	18.5 ± 0.2	21.6 ± 0.3	20.2 ± 0.0	20.4 ± 0.1	20.9 ± 0.2
	*p* value #	-------	0.002	0.017	0.008	0.001
*C. glabrata*	Mean ± SD	5.0 ± 0.0	18.8 ± 0.2	18.0 ± 0.1	19.0 ± 0.2	20.2 ± 0.2
	*p* value #	-------	1.5 × 10^−4^	8.7 × 10^−5^	1.4 × 10^−4^	1.2 × 10^−4^
*C. krusei*	Mean ± SD	0.0 ± 0.0	10.9 ± 0.1	9.1 ± 0.1	8.9 ± 0.1	11.1 ± 0.1
	*p* value #	-------	1.1 × 10^−4^	5.8 × 10^−5^	1.5 × 10^−4^	1.1 × 10^−4^

**#** Statistical analysis by One-Way ANOVA, Post hoc multiple Comparison, Dunnett T3 pattern against mean values of the Crude extract, number of replicates (*n* = 3), and Degree of freedom (df) = 2; SD: Standard Deviation.

**Table 4 molecules-27-03113-t004:** Zone of inhibition for bacterial strains treated with AgNPs of the Palm leaves aqueous extract.

Species		Non-Treated	AgNPs (25%)	AgNPs (50%)	AgNPs (100%)
*E. coli*	Mean ± SD	0.2 ± 0.1	13.3 ± 0.3	14.5 ± 0.1	14.8 ± 0.1
	*p* value #	-------	8.9 × 10^−5^	7.2 × 10^−8^	6.6 × 10^−8^
*B. subtilis*	Mean ± SD	0.1 ± 0.1	11.5 ± 0.5	12.2 ± 0.2	14.7 ± 0.4
	*p* value #	-------	0.002	6 × 10^−6^	4.5 × 10^−4^
*P. aeruginosa*	Mean ± SD	0.4 ± 0.2	14.0 ± 0.1	14.9 ± 0.1	15.3 ± 0.1
	*p* value #	-------	1.4 × 10^−5^	1.2 × 10^−5^	1.4 × 10^−4^
*S. aureus*	Mean ± SD	0.2 ± 0.1	18.8 ± 0.2	18.0 ± 0.1	19.0 ± 0.2
	*p* value #	-------	3.6 × 10^−7^	1.6 × 10^−8^	3.5 × 10^−7^

**#** Statistical analysis by One-Way ANOVA, Post hoc multiple Comparison, Dunnett T3 pattern against mean values of the Crude extract, number of replicates (*n* = 3), and Degree of freedom (df) = 2; SD: Standard Deviation.

**Table 5 molecules-27-03113-t005:** Zone of inhibition for bacterial strains treated with AgNPs of the Palm leaves ethanolic extract.

Species		Non-Treated	AgNPs (25%)	AgNPs (50%)	AgNPs (100%)
*E. coli*	Mean ± SD	0.2 ± 0.1	13.8 ± 0.1	14.1 ± 0.1	14.4 ± 0.1
	*p* value #	-------	8.7 × 10^−8^	6 × 10^−7^	2 × 10^−6^
*B. subtilis*	Mean ± SD	0.1 ± 0.1	12.0 ± 0.5	12.1 ± 0.1	13.9 ± 1.0
	*p* value #	-------	14.1 × 10^−5^	8.1 × 10^−8^	0.01
*P. aeruginosa*	Mean ± SD	0.4 ± 0.2	14.7 ± 0.2	15.1 ± 0.1	16.8 ± 0.2
	*p* value #	-------	9.5 × 10^−7^	1.1 × 10^−5^	3.9 × 10^−7^
*S. aureus*	Mean ± SD	0.2 ± 0.1	14.2 ± 0.1	14.2 ± 0.1	14.1 ± 0.1
	*p* value #	-------	7.8 × 10^−8^	7.7 × 10^−8^	7.9 × 10^−8^

**#** Statistical analysis by One-Way ANOVA, Post hoc multiple Comparison, Dunnett T3 pattern against mean values of the Crude extract, number of replicates (*n* = 3), and Degree of freedom (df) = 2; SD: Standard Deviation.

## Data Availability

All the data presented in this study are available within the current article. All statistical analysis results and raw data are available on request from the corresponding author.

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
