# Peer review of "Antimicrobial Activity of Green Silver Nanoparticles Synthesized by Different Extracts from the Leaves of Saudi Palm Tree (Phoenix Dactylifera L.)"

_molecules, 2022, doi:10.3390/molecules27103113_

Round 1

Reviewer 1 Report

The manuscript “Antimicrobial activity of Green Silver Nanoparticles synthesized by different extracts from the leaves of Saudi palm tree (Phoenix dactylifera L.)” assessed the antibacterial and antifungal properties of the silver nanoparticles synthesized from extracts of Phoenix dactylifera. The results showed that AgNPs had stronger antibacterial and antifungal activities than that of the crude extracts. The manuscript provides some interesting results. However, only crude extract was investigated. No chemical constituents were isolated from P. dactylifera, let alone the bioactive compounds. Only several chemical structures were proposed by GC/MS analysis. Therefore, the current version of the manuscript does not meet the typical standards for publication in the Molecules. A few additional items should be addressed before publication in any form:  

Line 122, the chromatogram of GC-MS analysis and the reason why the phenolics instead of others compounds (such as flavonoids) were measured should be provided.

Line 164, the concentrations or names of tested samples signed on the plates are too obscure in figure 4. 

Line 168, there are no obvious concentrations of aqueous, ethanolic extracts and AgNO3 on antifungal activity tests. 

There are some errors about Latin names, such as abbreviation of “Phoenix” (line 160, 189, 191 et al), the italic names, such as “C. albicans” (line 183), “E. coli” (line 238) et al. 

Line 199, why the two crude extracts and AgNO3 were not assessed on bactericidal activities same as antifungal activities?

Author Response

Reviewer #1

The manuscript “Antimicrobial activity of Green Silver Nanoparticles synthesized by different extracts from the leaves of Saudi palm tree (Phoenix dactylifera L.)” assessed the antibacterial and antifungal properties of the silver nanoparticles synthesized from extracts of Phoenix dactylifera. The results showed that AgNPs had stronger antibacterial and antifungal activities than that of the crude extracts. The manuscript provides some interesting results. However, only crude extract was investigated. No chemical constituents were isolated from P. dactylifera, let alone the bioactive compounds. Only several chemical structures were proposed by GC/MS analysis. Therefore, the current version of the manuscript does not meet the typical standards for publication in the Molecules. A few additional items should be addressed before publication in any form: 

Response: the currents study describes the effects of different extracts of Phoenix dactylifera L. all experiments were performed against aqueous and ethanolic extracts, in addition to AgNPs, were prepared from the crude extract of the leaves of Saudi palm tree (Phoenix dactylifera L.). this is similar tio many previous studies at natural antimicrobial agents such as (Rizwana, et al., 2021; Alamgir, 2017; Sani et al., 2017; Ansari et al., 2018; Abusahid et al., 2018; Al-zoreky et al., 2015) References [40-46]. All these studies used the crude extracts of different parts of palm plant to study their antimicrobial effects. Furthermore, the green synthesis of AgNPs relies fundamentally on the use of crude extracts of plant parts, as shown elsewhere [13-18]. However, many studies tested the antimicrobial activities of specific constituents from other plants, the current study aimed to show the effect of naturally prepared antimicrobial agents such as palm leaves, which might be medical importance because of higher safety levels. To clarify this point, we have added few changes in lines 27, 70-72, and 421-422.

The GC/MS analysis describes the raw data analysis of the tested materials, so the peaks resulted are the real-results. Authors can’t add more compounds, which are not resulted in the GC/MS analysis. Furthermore, similar previous studies showed similar GC/MS analysis of palm extract such as [30-34].

Line 122, the chromatogram of GC-MS analysis and the reason why the phenolics instead of others compounds (such as flavonoids) were measured should be provided.

Response: we agree with the reviewer comment. The products of palm trees had medical importance because of the high phenolic and flavonoids content, however, flavonoids are actually phenolic compounds but more volatile and thermolabile. Analysis of the non-volatile and thermolabile phenolic compounds by GC-MS presupposes their conversion into volatile and thermotolerant derivatives, which might increase the accuracy of the results as a result of increased stability. To clarify this point, we have added few changes in lines 146-148.

Line 164, the concentrations or names of tested samples signed on the plates are too obscure in figure 4.

Response: we agree with the reviewer comment. To clarify this point, we have added the concentrations of tested materials in the legends of figures, 4, 5, 6, 7 in lines 202, 203, 216, 220, 241, 243, 273, 298, 299.

Line 168, there are no obvious concentrations of aqueous, ethanolic extracts and AgNO3 on antifungal activity tests.

Response: we agree with the reviewer comment. To clarify this point, we have added the concentrations of tested materials in the legends of figures, 4, 5, 6, 7 and tables 2 and 3. Changes were made in lines 202, 203, 216, 220, 241, 243, 273, 298, 299.

There are some errors about Latin names, such as abbreviation of “Phoenix” (line 160, 189, 191 et al), the italic names, such as “C. albicans” (line 183), “E. coli” (line 238) et al.

Response: we agree with the reviewer comment. We have reviewed and corrects all Latin names as recommended.

Line 199, why the two crude extracts and AgNO3 were not assessed on bactericidal activities same as antifungal activities?

Response: we agree with the reviewer comment. As shown in tables 4 and 5. Neither the crude extracts nor the AgNO3 treatment-induced any inhibition of the bacterial growth (data not shown in the figures). We add this comment in line 255-257.

Reviewer 2 Report

The manuscript "Antimicrobial activity of Green Silver Nanoparticles synthesized by different extracts from the leaves of Saudi palm tree (Phoenix dactylifera L.)" is devoted to the synthesis of silver nanoparticles in an aqueous or ethanolic extract of Saudi palm tree and testing their antimicrobial activity. Biogenic synthesis of metal nanoparticles using extracts of different plants has been applied by many authors in order to obtain metal nanoparticles in a more environmentally-friendly way and to enhance the antimicrobial activity of the particles at the account of absorbed plant metabolites. The main advantage of the presented work is the use of a wide range of Candida species in antimicrobial tests. However, the synthesis of silver nanoparticles was not convincingly demonstrated by the authors (see the comments below). Considering that the silver nitrate solution also demonstrated an impressive antimicrobial activity, comparable to that of the silver nanoparticles, robust confirmation of the nanoparticle synthesis is of utmost importance.

  1. The authors should measure the concentration of obtained nanoparticles (in µg/ml), for example by evaporating and weighing a certain volume of the nanoparticle solution. It is very important to know the exact concentration of nanoparticles in order to compare the antimicrobial efficacy of the Saudi palm tree-derived silver nanoparticles with the efficacy of silver nanoparticles obtained by other ways or using other plant extracts.
  2. Why did the authors stain their TEM preparations? Silver nanoparticles are perfectly visible in TEM without staining, while staining of all preparations with uranyl acetate and lead citrate made it impossible to see the difference between Fig 2A, Fig2B and Fig2C and assess the efficiency of the nanoparticle synthesis. Moreover, what are the large particles visible at the bottom of Fig. 2A, which seem to have the same sizes as some of the particles in Fig. 2B and Fig. 2C?
  3. How were the solutions prepared for UV-vis measurements? As it can be seen in Fig. 1B the spectrum of biosynthesized AgNP contained all the same peaks as the initial plant extract. The only (and surprising) difference was a lower overall absorption of the AgNP solution compared to the initial plant extract. I guess that this could be due to the AgNP solution dilution in the course of chromatography. To avoid this, the authors should have measured the AgNP solution against the blank solution of the plant extract that had been undergone all the same treatments as the AgNP solution, including chromatography. Moreover, the appearance of AgNP solution spectrum is not characteristic of AgNP solution spectrum published by other authors, where a distinct peak was observed between 400-500 nm. Again, the problem could be resolved by using an appropriate blank solution.
  4. Please, present all images in Fig. 5 at the same magnification and add an image demonstrating several control cells. One control cell is not representative of the whole control cell population. What do the arrows in the control image point to?
  5. Please, present all images in Fig. 6 at the same magnification. What do the arrows in the control image point to?
  6. Other possibilities of biological synthesis of AgNP should be mentioned in the Introduction, such as microbe- or algae- assisted synthesis (see for example: https://doi.org/10.1016/j.tibtech.2016.02.006, https://doi.org/10.3390/mi12121480).

Author Response

Reviewer #2

The manuscript "Antimicrobial activity of Green Silver Nanoparticles synthesized by different extracts from the leaves of Saudi palm tree (Phoenix dactylifera L.)" is devoted to the synthesis of silver nanoparticles in an aqueous or ethanolic extract of Saudi palm tree and testing their antimicrobial activity. Biogenic synthesis of metal nanoparticles using extracts of different plants has been applied by many authors in order to obtain metal nanoparticles in a more environmentally-friendly way and to enhance the antimicrobial activity of the particles at the account of absorbed plant metabolites. The main advantage of the presented work is the use of a wide range of Candida species in antimicrobial tests. However, the synthesis of silver nanoparticles was not convincingly demonstrated by the authors (see the comments below). Considering that the silver nitrate solution also demonstrated an impressive antimicrobial activity, comparable to that of the silver nanoparticles, robust confirmation of the nanoparticle synthesis is of utmost importance.

  1. The authors should measure the concentration of obtained nanoparticles (in µg/ml), for example by evaporating and weighing a certain volume of the nanoparticle solution. It is very important to know the exact concentration of nanoparticles in order to compare the antimicrobial efficacy of the Saudi palm tree-derived silver nanoparticles with the efficacy of silver nanoparticles obtained by other ways or using other plant extracts.

Response: we agree with the reviewer comment. In the methodology section line 332-333, we mentioned that the synthesized particles were purified by the Size Exclusion Chromatography (SEC) method. In that method we obtained a powder form AgNPs, then we prepared a concentration of 10 mg/ml that have been used in different experiments. To clarify this point, we have added the concentrations of tested materials in the legends of figures, 4, 5, 6, 7 in lines 202, 203, 216, 220, 241, 243, 273, 298, 299.

  1. Why did the authors stain their TEM preparations? Silver nanoparticles are perfectly visible in TEM without staining, while staining of all preparations with uranyl acetate and lead citrate made it impossible to see the difference between Fig 2A, Fig2B and Fig2C and assess the efficiency of the nanoparticle synthesis. Moreover, what are the large particles visible at the bottom of Fig. 2A, which seem to have the same sizes as some of the particles in Fig. 2B and Fig. 2C?

Response: we agree with the reviewer comment. The TEM analysis was used only to detect the size range of the synthetized nanoparticles as compared to the crude extract. For accurate comparison, we have to stain all slides with uranyl acetate and lead citrate to increase the contrast of the ultrastructure. This is in agreement with the method described by Chen et al., 2017 (see below). To clarify this point, we add few changes to the result section line 125-127. The unclear large particles at the bottom of Fig. 2A are artificial background that resulted from the preparation method, so it can be ignored.

Chen, S., Goode, A. E., Skepper, J. N., Thorley, A. J., Seiffert, J. M., Chung, K. F., Tetley, T. D., Shaffer, M. S., Ryan, M. P., & Porter, A. E. (2016). Avoiding artefacts during electron microscopy of silver nanomaterials exposed to biological environments. Journal of microscopy261(2), 157–166. https://doi.org/10.1111/jmi.12215

  1. How were the solutions prepared for UV-vis measurements? As it can be seen in Fig. 1B the spectrum of biosynthesized AgNP contained all the same peaks as the initial plant extract. The only (and surprising) difference was a lower overall absorption of the AgNP solution compared to the initial plant extract. I guess that this could be due to the AgNP solution dilution in the course of chromatography. To avoid this, the authors should have measured the AgNP solution against the blank solution of the plant extract that had been undergone all the same treatments as the AgNP solution, including chromatography. Moreover, the appearance of AgNP solution spectrum is not characteristic of AgNP solution spectrum published by other authors, where a distinct peak was observed between 400-500 nm. Again, the problem could be resolved by using an appropriate blank solution.

Response: we agree with the reviewer comment. The solutions for UV-vis measurements were prepared as been described by Pingale et al., 2018. The purified AgNPs were objected to 200–800 nm UV light by UV-2450 double-beam according to the manufacturer instruction, and scanning interval was 0.5 nm. Here we tested both preparations, purified by the synthesized particles were purified by the Size Exclusion Chromatography (SEC) method, versus the crude aqueous and ethanolic extracts of palm leaves. To be honest we couldn’t purify the crude extracts of palm leaves, as we tried to keep as much as possible of its natural constituents. However, for the spectrum for AgNPs of the aqueous extract showed higher broad peak at 400 nm, the peaks of ethanolic extract produced three sharp absorption peaks at 300, 429, and 700 nm, besides, one broad beak at 200-250 nm. That indicated the formation of AgNPs with larger sizes that were, unlike water AgNPs, not dispersed. This is in agreement with a previous study of Ashraf et al., 2016 that showed that AgNPs (20 μM) were recorded in the wavelength range of 200–250 nm, while in our study AgNPs appeared at little higher peaks at 300 nm. We have added few changes at lines 112-113. We have added the reference of Ashraf et al., 2016.

  1. Please, present all images in Fig. 5 at the same magnification and add an image demonstrating several control cells. One control cell is not representative of the whole control cell population. What do the arrows in the control image point to?

Response: we agree with the reviewer comment. Fig. 5B and 5C showed the morphological changes induced to C. albicans upon treatment with AgNPs of the either ethanolic or aqueous extracts. As seen, the treated fungus was smaller in size than the untreated control. We used higher magnification to show more cellular details indicated by the arrows. The arrows in control image indicated the normal morphology of C. albicans such as the cell wall and intracellular organelles. In Fig. 5B and 5C, the arrows indicated the rapture of the cellular membrane, the shrinkage of the cells, without clear distinguishing of the cellular organelles. We have updated this section in lines, 227-232.

  1. Please, present all images in Fig. 6 at the same magnification. What do the arrows in the control image point to?

Response: we think that the reviewer meant Fig. 7. Again, we used higher magnification to show more cellular details indicated by the arrows. The arrows in control image indicated the normal morphology of P. aeruginosa such as the cell wall and intracellular organelles. In Fig. 7B and 7C, the arrows indicated the rapture of the cellular membrane (pale color) and the damage of the cellular organelles. We have updated this section in lines, 293-297.

  1. Other possibilities of biological synthesis of AgNP should be mentioned in the Introduction, such as microbe- or algae- assisted synthesis (see for example: https://doi.org/10.1016/j.tibtech.2016.02.006, https://doi.org/10.3390/mi12121480).

Response: we agree with the reviewer's comment. We have updated the introduction and references sections as recommended.

Round 2

Reviewer 2 Report

Unfortunately, I am not satisfied with the authors' replies to some of my comments

  1. The authors have written that the concentration of obtained AgNP was 10 mg/ml (V/V). What does (V/V) mean?
  2. As concerns my comment 2: I cannot agree with the authors that "The unclear large particles at the bottom of Fig. 2A are artificial background that resulted from the preparation method, so it can be ignored". These large particles look the same as those in Fig 2B and Fig 2C. Then, maybe everything that one can see in Fig 2B and Fig 2C is also an artificial background? I cannot see the real differences between the three images presented in Fig 2.
  3. As concerns my comment 3: The authors reply that: "However, for the spectrum for AgNPs of the aqueous extract showed higher broad peak at 400 nm, the peaks of ethanolic extract produced three sharp absorption peaks at 300, 429, and 700 nm, besides, one broad beak at 200-250 nm." But as one can see from Fig. 1B, all cited absorption peaks at 300, 429, and 700 nm, as well as a broad peak at 200-250 nm were also present in the spectrum of ethanolic plant extract, so there is no evidence of AgNP formation according to results of UV-Vis absorption, at least for ethanolic plant extract.
  4. My comment about presenting more than one control cell in Fig. 5 to demonstrate the morphology of control cells was completely ignored. I still think that one cell cannot represent the whole cell population. The magnification of images in Fig. 5 and in Fig 7 was also not changed. The authors reply that "We used higher magnification to show more cellular details indicated by the arrows". But it is difficult to compare cellular details as well as cell sizes between control and treated cells when they have different magnifications.
  5. The authors have added to the Conclusion that "The antimicrobial efficacy of AgNPs synthesized from natural crude extracts from palm leaves, might suggest its medical applications because of high safety levels."(Lines 529-531). However, the safety of the synthesized nanoparticles was not tested at all, so, it is not possible to make such a statement in the Conclusion.

Author Response

Reviewer #2

  1. The authors have written that the concentration of obtained AgNP was 10 mg/ml (V/V). What does (V/V) mean?

Response: we agree with the reviewer comment that this might cause some confusion to the readers. We used a stock solution of 10mg/ml concentration for all AgNPs prepared but the working solutions (doses) were 25, 50, and 100% at which we mixed the AgNPS with 75, 50, and 0% of the growth media, respectively. SO (V/V) mean the volume ratio between the AgNPs and Culture media. To clarify this point, we have deleted (V/V) term.

  1. As concerns my comment 2: I cannot agree with the authors that "The unclear large particles at the bottom of Fig. 2A are artificial background that resulted from the preparation method, so it can be ignored". These large particles look the same as those in Fig 2B and Fig 2C. Then, maybe everything that one can see in Fig 2B and Fig 2C is also an artificial background? I cannot see the real differences between the three images presented in Fig 2.

Response: we agree with the reviewer comment that this might cause some confusion to the readers. We have prepared and attached another figure that shows clear TEM images for AgNO3 particles with the 150000× magnification. The new figure showed the size of AgNO3, Aqueous of extract Phoenix dactylifera L leaves, and AgNPs of Phoenix dactylifera L leaves aqueous extract.

  1. As concerns my comment 3: The authors reply that: "However, for the spectrum for AgNPs of the aqueous extract showed higher broad peak at 400 nm, the peaks of ethanolic extract produced three sharp absorption peaks at 300, 429, and 700 nm, besides, one broad beak at 200-250 nm." But as one can see from Fig. 1B, all cited absorption peaks at 300, 429, and 700 nm, as well as a broad peak at 200-250 nm were also present in the spectrum of ethanolic plant extract, so there is no evidence of AgNP formation according to results of UV-Vis absorption, at least for ethanolic plant extract.

Response: we agree with the reviewer comment that this might cause some confusion to the readers. We have found that we by mistake replaced the data of the crude extract with that of AgNPs for ethanol result. We have prepared and attached another figure that shows the correct UV results. From the new attached figure, we found that AgNPs had higher absorbance values than the crude extract, which proves the formation of AgNPs.

  1. My comment about presenting more than one control cell in Fig. 5 to demonstrate the morphology of control cells was completely ignored. I still think that one cell cannot represent the whole cell population. The magnification of images in Fig. 5 and in Fig 7 was also not changed. The authors reply that "We used higher magnification to show more cellular details indicated by the arrows". But it is difficult to compare cellular details as well as cell sizes between control and treated cells when they have different magnifications.

Response: we agree with the reviewer comment and we regret to disappoint his expectations. We have attached another figure at the same magnification (60000×) the candida without treatment.

  1. The authors have added to the Conclusion that "The antimicrobial efficacy of AgNPs synthesized from natural crude extracts from palm leaves, might suggest its medical applications because of high safety levels."(Lines 529-531). However, the safety of the synthesized nanoparticles was not tested at all, so, it is not possible to make such a statement in the Conclusion.

Response: we agree with the reviewer's comment. We have removed such stamen from the conclusion.
